# Genome-Wide and Exome-Wide Association Study Identifies Genetic Underpinning of Comorbidity between Myocardial Infarction and Severe Mental Disorders [note 1]

**DOI:** 10.3390/biomedicines12102298

**Published:** 2024-10-10

**Authors:** Bixuan Jiang, Xiangyi Li, Mo Li, Wei Zhou, Mingzhe Zhao, Hao Wu, Na Zhang, Lu Shen, Chunling Wan, Lin He, Cong Huai, Shengying Qin

**Affiliations:** 1Bio-X Institutes, Key Laboratory for the Genetics of Developmental and Neuropsychiatric Disorders (Ministry of Education), Shanghai Jiao Tong University, Shanghai 200030, China; bixuanjiang@sjtu.edu.cn (B.J.); to_lixiangyi@sjtu.edu.cn (X.L.); oxygen26@163.com (H.W.); zhangna_1991@126.com (N.Z.); yoyomailer@sjtu.edu.cn (L.S.); clwan@sjtu.edu.cn (C.W.); helin@sjtu.edu.cn (L.H.); 2Department of Cardiology of The Second Affiliated Hospital, School of Medicine, Zhejiang University, Hangzhou 310009, China; limo1169718691@163.com; 3State Key Laboratory of Transvascular Implantation Devices, Hangzhou 310009, China; 4Cardiovascular Key Laboratory of Zhejiang Province, Hangzhou 310009, China; 5Ministry of Education—Shanghai Key Laboratory of Children’s Environmental Health & Department of Developmental and Behavioural Paediatric & Child Primary Care, Xinhua Hospital Affiliated to Shanghai Jiao Tong University School of Medicine, Shanghai 200092, China; weizhousjtu@sjtu.edu.cn; 6Affiliated Mental Health Center & Hangzhou Seventh People’s Hospital, Zhejiang University School of Medicine, Hangzhou 310013, China; zhaomingzhe@zju.edu.cn; 7Sichuan Research Institute, Shanghai Jiao Tong University, Chengdu 610213, China

**Keywords:** comorbidity, mental disorder, myocardial infarction, GWAS, EWAS, association study

## Abstract

Background: Myocardial Infarction (MI) and severe mental disorders (SMDs) are two types of highly prevalent and complex disorders and seem to have a relatively high possibility of mortality. However, the contributions of common and rare genetic variants to their comorbidity arestill unclear. Methods: We conducted a combined genome-wide association study (GWAS) and exome-wide association study (EWAS) approach. Results: Using gene-based and gene-set association analyses based on the results of GWAS, we found the common genetic underpinnings of nine genes (*GIGYF2*, *KCNJ13*, *PCCB*, *STAG1*, *HLA-C*, *HLA-B*, *FURIN*, *FES*, and *SMG6*) and nine pathways significantly shared between MI and SMDs. Through Mendelian randomization analysis, we found that twenty-seven genes were potential causal genes for SMDs and MI. Based on the exome sequencing data of MI and SMDs patients from the UK Biobank, we found that *MUC2* was exome-wide significant in the two diseases. The gene-set analyses of the exome-wide association study indicated that pathways related to insulin processing androgen catabolic process and angiotensin receptor binding may be involved in the comorbidity between SMDs and MI. We also found that six candidate genes were reported to interact with known therapeutic drugs based on the drug–gene interaction information in DGIdb. Conclusions: Altogether, this study revealed the overlap of common and rare genetic underpinning between SMDs and MI and may provide useful insights for their mechanism study and therapeutic investigations.

## 1. Introduction

Cardiovascular diseases (CVDs), affecting the heart and blood vessels, are the primary cause of mortality worldwide. In recent decades, the global burden of CVDs has significantly increased. From 1990 to 2019, the number of people affected by CVDs nearly doubled, rising from an estimated 271 million to approximately 523 million, and the number of CVD deaths increased from 12.1 million to 18.6 million [1]. Myocardial Infarction (MI) is one of the most severe and life-threatening manifestations of CVDs, which is defined as myocardial cell death due to prolonged ischemia [2,3].

Severe mental disorders (SMDs), such as schizophrenia (SCZ) and bipolar disorder (BD), have affected approximately 1% of the UK population. SCZ and BD are two typical types of SMDs and share a common genetic cause of mainly additive genetic effects [4]. People with mental disorders are at an almost four times higher risk of death than those without mental disorders, and life expectancy in SCZ patients is approximately 15 to 20 years shorter compared to the general population [5,6].

Several intriguing relationships have been observed between SMDs and MI. For example, the high mortality among patients with SMDs is attributable to heart disease. In a U.S. national study of adult patients with schizophrenia aged 20 to 64, CVDs were identified as the leading cause of death, with a mortality rate of 403.2 per 100,000 person-years [7]. Similarly, an eight-year follow-up study of individuals with severe mental illness found that the mortality rates from respiratory and cardiovascular conditions were up to four times higher than those in the general population [8]. Additionally, research has shown that MI patients are at a higher risk of developing bipolar disorder [9]. Several environmental and lifestyle factors contribute to the increased risk of both MI and SMDs. These include tobacco use, inadequate physical activity, limited access to healthcare, and the metabolic side effects of antipsychotic medications [10]. Various biological mechanisms have been proposed to explain the complex interplay between mental health and cardiovascular health, including the dysfunction of the autonomic nervous system, the dysregulation of the hypothalamic–pituitary–adrenal axis, chronic inflammation, imbalanced neurotransmitters, and increased platelet reactivity [11,12,13,14].

From a genetic perspective, pleiotropy occurs when one gene affects multiple phenotypes. Meta-analyses of GWASs can elucidate the regions of the genome that associate with a disease, quantitative trait, or biomarker by tagging and testing the association of single-nucleotide polymorphisms (SNPs) in certain populations. Sivakumaran et al. conducted a systematic evaluation of pleiotropy among SNPs and genes recorded in the National Human Genome Research Institute’s Catalog and found abundant evidence of pleiotropy, specifically, that 16.9% of the genes and 4.6% of the SNPs have pleiotropic effects on common complex diseases and traits [15]. So et al. applied Mendelian randomization and polygenic risk scores to investigate the shared genetic basis of cardiovascular and metabolic effects of SCZ and BD, which showed that patients with SCZ might be genetically predisposed to cardiometabolic abnormalities, although this genetic predisposition was not observed in patients with BD [16]. Rødevand et al. revealed a polygenic overlap of 163 distinct common variants shared between loneliness, SMDs, and CVD risk factors, suggesting that the genetic basis of loneliness may increase the risk of both SMDs and CVD [17].

For most traits, even the most important loci identified by GWASs have small effect sizes and only explain a modest fraction of the predicted genetic variance, which is referred to as the mystery of the “missing heritability” [18,19]. Population-based whole-exome sequencing (WES) can discover rare risk alleles and complement established gene-mapping paradigms [20]. Rare coding variants with lower allele frequencies but larger effect sizes also contribute to genetic variance, especially for diseases with complex etiology such as SCZ [20,21]. However, to the best of our knowledge, the potential common and rare genetic underpinning of comorbidity between MI and SMDs remains unexplored.

In this study, taking SCZ and BD as representatives of SMDs, we investigated the potentially shared rare and common genetic underpinnings, including genes and pathways, that might contribute to the comorbidity relationship between MI and SMDs. We used both meta-analysis GWAS and discovery WES approaches to provide additional insights into the role of genes and pathways with pleiotropic effects on MI and SMDs. We hope that our investigation can help reveal the potential pleiotropy that might have implications for understanding the comorbidity of MI and SMDs and finding some candidate drug targets.

## 2. Materials and Methods

### 2.1. GWAS Summary Statistics

We used the publicly available GWAS summary statistics from the Psychiatric Genomics Consortium (PGC, https://pgc.unc.edu/for-researchers/download-results/, accessed on 29 August 2023) and the Cardiovascular Disease Knowledge Portal (CVDKP, https://cvd.hugeamp.org/, accessed on 29 August 2023). Basic information on the cohorts is listed in Table 1, and clinical information on cases and controls was in the original publications [22,23,24]. All samples in the cohorts of GWAS summary statistics were individuals of European ancestry.

### 2.2. Gene-Based and Gene-Set Analyses for Common Variants

We applied the Multi-marker Analysis of GenoMic Annotation (MAGMA version 1.10) approach to discover the genes and gene sets associated with each disorder [25]. The summary statistics, including the *p* value and SNP IDs, were as the input for MAGMA.

We annotated SNPs to genes based on their NCBI 37.3 genomic location (downloaded from http://ctglab.nl/software/magma, accessed on 13 September 2022) by using the 1000 Genomes European reference panel. To include regulatory elements for further analyses, we extended a 5 kb upstream and downstream window around each gene. Then, the gene-based analyses were conducted based on the annotation results by using the SNP-wise mean model.

For gene-set analyses, we collected pathways including GO Biological Process (GOBP), GO Cellular Component (GOCC), and the Kyoto Encyclopedia of Genes and Genomes (KEGG) from the Human Molecular Signatures Database (MSigDB, https://www.gsea-msigdb.org/gsea/msigdb, accessed on 26 July 2023), version 2023.1.Hs. The gene *p* values from the gene-based analyses was converted to Z-scores, which were used to assess whether genes in the gene sets above were associated with a given phenotype compared with other genes not in the gene sets [25].

We manually investigated the shared significant genes between MI and SMDs in the International Mouse Phenotyping Consortium (https://www.mousephenotype.org/, accessed on 20 March 2024) to discover their significant phenotypes in knockout mice.

### 2.3. The Potential Causal Genes for Myocardial Infarction and Severe Mental Disorders

For SCZ, BD, and MI, we used the driver-tissue estimation by selective expression to predict the most relevant tissue. Default parameters were employed except for statistical adjustments, where we implemented the Benjamini–Hochberg procedure to control false discovery rate (FDR) at a *p* value threshold of 0.05 for MI and the standard Bonferroni correction to conservatively maintain a Family-Wise Error Rate below the *p* value threshold of 0.05 for SCZ and BD. We found that Artery-Tibial was the most significant tissue for MI and Brain Frontal Cortex (BA9) was the most significant tissue for SCZ and BD (Appendix A). Then, we used the effective-median-based Mendelian randomization framework for inferring the causal genes of complex phenotypes (EMIC) to assess the candidate causal genes based on the GWAS summary results of MI, SCZ, and BD and the expression quantitative trait loci (eQTL) summary statistics of Artery-Tibial and Brain Frontal Cortex (BA9) which were downloaded from https://mailsysueducn-my.sharepoint.com/:f:/g/personal/limiaoxin_mail_sysu_edu_cn/EnhWhqLUNcpOrh6O3enFvCUBRvQ13v2970tcpOnNmmlKyg?e=1jkl06, (accessed on 2 February 2024), calculated by Li et al. based on the GTEx database (v8) [26,27]. The EUR panel (Phase 3) of the 1000 Genomes Project was used to generate LD matrices for EMIC. Only eQTLs with a *p* value less than 1 × 10^−6^ were included in the MR analyses. Genes with an EMIC *p* value lower than 0.05 were used to further perform the EMIC pleiotropy fine-mapping analyses. Other parameters were default.

### 2.4. Exome Sequencing in UK Biobank and Case–Control Ascertainment

UK Biobank (UKBB) is a large population-based cohort that collected approximately 500,000 individuals aged 40–69 years at enrollment from the United Kingdom with phenotype data and genotype data. Here, we conducted analyses for ~200,000 individual WES data released in October 2020 from UKBB under application number 34716. All participants had granted written informed consent for their participation.

In the UKBB 200k exome cohort, 590, 322, and 3673 individuals were classified as BD with F31 (mean age 55.5 years), SCZ with F20 (mean age 54.0 years), and MI with I21 (mean age 61.7 years), according to the International Classification of Diseases version-10 (ICD-10, UKBB Data-Field 41202). In addition, 2000 randomly selected individuals without any mental and behavioral disorder, nervous system disease, or circulatory system disease diagnoses (F00-F99, G00-G99, or I00-I99) were defined as control (mean age 54.8 years) for the next step analyses. Demographic and clinical characteristics of all UKBB samples, cases, and controls in this study are listed in Table 2. Cases and controls of European ancestry were retained for further study.

Due to the UKBB 200k exome cohort lacking essential quality control and filtering, we applied several quality control steps on variants and samples. On the variant-level quality control, we removed SNPs with missing genotype rate > 5% and Hardy–Weinberg equilibrium test (*p* value  <  1 × 10^−6^). On the sample-level quality control, individuals with non-European ethnic backgrounds (Data-Field 21000), a missing genotype rate > 5%, and a heterozygosity rate > 3*sigma were removed from further analyses for reasons of genetic outliers or blood relationship. Variant- and sample-level quality control steps were performed using the Variant Quality Score Recalibration of Genotype Analysis ToolKit (GATK, version 4.2, https://software.broadinstitute.org/gatk/, accessed on 19 February 2021), PLINK (version 1.9, http://www.cog-genomics.org/plink, accessed on 13 August 2023), and R (version 4.3.2) [28,29,30].

### 2.5. Exome-Wide Association Study

To identify genes and pathways associated with each trait, we analyzed the combined effects of rare missense or loss of function (including frameshift, stop-gain, and splicing) variants. Variants with a minor allele frequency lower than 0.01 in European ancestry public databases, including the Genome Aggregation Database (gnomAD, exome and genome) and the 1000 Genomes Project, were defined as rare variants. The filtration, annotation, and statistical tests of the gene-based and gene-set association analyses for rare variants were performed using KGGSeq [31].

Genes with at least 2 rare missense or loss of function variants were analyzed using the optimal sequencing kernel association test (SKAT-O), which combines burden test and SKAT and provides high power for variants that had different directions of effects on increasing disease risk or protecting from disease [32].

In the gene-set association analyses, we performed both the Combined Multivariate and Collapsing (CMC) burden test and the SKAT-O approach for each of the case–control groups in BD, SCZ, and MI, separately on canonical pathways, GOBP, GOCC, and GO Molecular Function (GOMF).

### 2.6. Prediction of Drug-Gene Interaction

The Drug–Gene Interaction Database (DGIdb v4.2.0, https://dgidb.org/, accessed on 27 February 2024) is a public database containing more than 10,000 genes and 15,000 drugs involved in over 50,000 drug–gene interactions [33]. The shared genes identified from GWAS analyses, MR analyses, and WES analyses were considered as potential pharmaceutical targets and submitted to DGIdb to explore existing drugs for the potential treatment of SMD-MI comorbidity. Interactions that were supported by one or more reliable publication(s) were considered trustworthy. The relationships between targets, drugs, and disorders were visualized using Cytoscape 3.10.2.

### 2.7. Statistical Analysis and Significance Levels

In GWAS analyses, we used Bonferroni correction for multiple testing in gene-based analyses, setting the significance threshold at 0.05/n, where n is the number of genes with valid SNPs, and nominal *p* values below 0.05 and Benjamini–Hochberg FDR values below 0.25 were considered significant for gene-set analyses. In exome-wide association studies, we applied the SKAT-O test for rare variants, with a nominal *p* value threshold of 0.05 and an FDR less than 0.25 for the gene-level test, and the threshold for statistical significance was set at a nominal *p* value of 0.05 for both CMC and SKAT-O tests for the gene-set association test. For causal gene prediction, we used the EMIC framework, considering genes with *p* values < 0.05 as candidate causal genes. All analyses were conducted using MAGMA (version 1.10), KGGSeq (V1.0+), GATK (version 4.2), PLINK (version 1.9), and R (version 4.3.2).

## 3. Results

### 3.1. Potential Pleiotropic Genes for Myocardial Infarction and Severe Mental Disorders in GWAS Analyses

We used MAGMA to determine the combined effect of common genetic variants within genes. There were 18,206 genes in MI, and 18,360 and 18,358 genes in SCZ and BD, respectively. Through MAGMA gene-based analyses, 146 genes were MI-related significant genes; 619 genes and 191 genes were SCZ-related and BD-related significant genes (*p* value < alpha/number of genes tested; alpha = 0.05). We identified nine pleiotropic genes shared between SCZ and MI, and three pleiotropic genes shared between MI and BD. Notably, among those genes, three genes (*HLA-C*, *HLA-B*, and *FURIN*) were significant in SCZ, BD, and MI. The detailed information of these pleiotropic genes is listed in Table 3.

### 3.2. Shared Pathways of Myocardial Infarction and Severe Mental Disorders by Gene-Set Analyses for Common Architecture

We integrated the gene-level analysis results from three different diseases or disorders to elucidate the common functional pathways shared between MI and SMDs. We conducted gene-set association analyses using GOCC, GOBP, and KEGG pathways as predefined gene sets in MAGMA. Gene-set analyses identified seven significant gene sets shared between BD and MI, and two significant gene sets shared between SCZ and MI, each achieving a *p* value less than 0.05 and a FDR below 0.25 (Figure 1).

Multiple shared gene sets, including GOBP regulation of catecholamine uptake involved in synaptic transmission (*p* value = 5.26 × 10^−4^ for SCZ, *p* value = 1.38 × 10^−2^ for MI), GOBP neurotransmitter transport (*p* value = 9.90 × 10^−4^ for BD, *p* value = 1.56 × 10^−2^ for MI), and GOBP regulation of neurotransmitter levels (*p* value = 8.73 × 10^−4^ for BD, *p* value = 8.23 × 10^−3^ for MI) that achieved statistical significance, revealed the importance of neurotransmitters in the shared genetic basis of MI and SMDs.

The GOBP dendrite arborization gene set, consisting of ten genes, was the top-scoring GOBP shared pathway in both SCZ and MI gene sets. This gene set identified Neuropilin 1 (*NRP1*) as significantly associated with SCZ and MI (*p* value = 1.18 × 10^−5^ for SCZ; *p* value = 1.04 × 10^−3^ for MI; gene analyses in MAGMA).

### 3.3. Shared Causal Genes for Myocardial Infarction and Severe Mental Disorders in Mendelian Randomization Analyses

We further estimated the potential causal genes through Mendelian randomization analyses. Using GWAS summary statistics for SMDs and MI, along with gene-level eQTLs, the MR analyses detected 127, 165, and 288 nominally significant causal genes (*p* value < 0.05). Among these, nine genes (*RNF5P1*, *SERBP1P3*, *HLA-DQB1-AS1*, *CYP21A1P*, *HLA-C*, *MTRF1L*, *HLA-DQA2*, *C15orf40*, and *KANSL1-AS1*) displayed consistent significance across BD, SCZ, and MI (Appendix A). Of particular note, *HLA-C* (*p* value = 0.0433 for BD, 3.76 × 10^−4^ for SCZ, and 4.16 × 10^−4^ for MI) was shared in both MR analyses and GWAS gene-based analyses (Table 3). Additionally, the analyses revealed that four genes (*CENPQ*, *STK19B*, *HLA-DRB6*, and *GSTM3*) and fourteen genes (*ERAP2*, *NBR2*, *TRMT61B*, *LRRC37A4P*, *WFDC3*, *SFTA1P*, *MAPK8IP1P2*, *HLA-DOB*, *NPIPB2*, *DND1P1*, *AMH*, *LRRC37A2*, *ZNF589*, and *LRRC37A*) were associated with a potentially increased risk of comorbidity between BD and MI (Appendix A) and between SCZ and MI (Appendix A), respectively.

### 3.4. Potential Pleiotropic Genes for Myocardial Infarction and Severe Mental Disorders in WES Analyses

To find the rare shared architecture of MI and SMDs, we further examined the rare functional variants within WES data of approximately 200,000 participants in UKBB and then performed gene-based exome-wide association analyses. After sample-level quality control, a total of 540 European cases for BD, 254 for SCZ, and 3401 for MI were retained for further analysis.

Genes containing more than two rare missense or loss of function variants were tested in the gene-based association by RVtests. Mucin 2, oligomeric mucus/gel-forming (*MUC2*), presented significance in both SCZ (*p* value = 0.039; FDR = 0.249) and MI (*p* value = 9.01 × 10^−5^; FDR = 0.0492) (Figure 2).

### 3.5. Shared Pathways of Myocardial Infarction and Severe Mental Disorders by Gene-Set Analyses for Rare Architecture

To identify the underlying shared pathways associated with MI and SMDs in WES analyses, we performed gene-set based analyses using a list of predefined candidate pathways. Gene sets with both CMC and SKAT-O *p* values less than 0.05 were viewed as significant related to the certain disorder.

Shared significant canonical pathways included mammary gland development pathway embryonic development stage 1 of 4 between BD and MI, and RIP-mediated NFkB activation via ZBP1 between SCZ and MI. When using GOBP as input gene sets for RVtests, all shared significant biological processes consistently achieved statistical significance across two tests including insulin processing (in BD and MI), neutrophil-mediated killing of Gram-negative bacterium (in SCZ and MI), and androgen catabolic process (in BD, SCZ, and MI). Only one shared GOMF gene-set showed significance between MI and SMDs (angiotensin receptor binding in BD and MI). No GOCC gene set achieved shared significance between BD and MI or between SCZ and MI. Further information about the gene-set analysis results of WES is listed in Appendix A.

### 3.6. Potential Therapeutic Drugs from Gene–Drug Interaction Prediction

The drug–gene interaction results from DGIdb indicated that 28 drugs or compounds interacted with the six shared genes of BD, SCZ, and MI (Figure 3). Multiple drugs can target the same gene for their effects (such as 16 drugs targeting *HLA-B*, 10 drugs targeting *HLA-C*, and 3 drugs targeting *NRP1*).

We found that 21 drugs interacted with *HLA-B* and *HLA-C*, which were the shared genes of BD, SCZ, and MI. One antipsychotic (i.e., olanzapine) had been reported to interact with *GSTM3*, which was shared by BD and MI.

## 4. Discussion

Herein, we separately applied gene-based and gene-set association analyses on GWAS and WES data to discover the shared common and rare genetic underpinnings including genes and pathways between MI and SMDs. Through meta-analyses of the largest GWASs to our knowledge, we identified nine genes shared between SCZ and MI, three genes shared between BD and MI, and nine pathways significantly shared between these two types of disorders. Notably, three genes (*HLA-C*, *HLA-B*, and *FURIN*) were found to have a joint effect on BD, SCZ, and MI. The association analyses on WES from the UKBB cohort revealed one additional gene (*MUC2*) showing exome-wide significance in SCZ and MI and several gene sets shared by both types of disorders.

Through genome-wide and exome-wide association studies, our results support the hypothesis that both rare and common genetic underpinnings, including specific genes and pathways, can mediate the interplay between SMDs and MI. This finding suggests that integrating rare functional variants, which may have fewer carriers but larger effect sizes, with common variants can provide a more comprehensive understanding of the genetic mechanisms underlying the comorbidity of these conditions. By combining the effects of both rare and common genetic variations, we can better explain the complex interplay between SMDs and MI, leading to a more complete picture of the genetic underpinnings involved in their co-occurrence.

We used MAGMA [25] to reveal common architecture overlap between SMDs and MI. Grb10-interacting GYF protein 2 (*GIGYF2*), as an RNA-binding protein promotes endothelial cell senescence, dysfunction, and inflammation and has high expression in senescent human endothelial cells and the aortas of aged mice [34]. *GIGYF2* also implicates the translational control process in SCZ by a zebrafish mutant ortholog study of SCZ-associated genes focusing on abnormal behavior and brain activity [35]. FES Upstream Region (*FURIN*) is located at an open reading frame upstream of FES Proto-Oncogene, Tyrosine Kinase (*FES*). *FURIN* may have opposing effects in the pathogenesis of SCZ and MI. The inhibition of FURIN in mice results in the prevention of vascular MMP2 activation and a decrease in vascular remodeling and may attenuate the development of CVDs [36]. *FES* exerts a protective effect against CVDs by a genome editing study and suggests that enhanced FES activity may be a potential new treatment for CVDs intervention [37]. The attenuated expression of *furin1*, a *FURIN* homolog in *Drosophila*, presents defective habituation to recurrent footshocks, which constitutes a SCZ endophenotype [38]. SMG6 nonsense-mediated mRNA decay factor (*SMG6*) also impacts myocardial injury and neurogenesis defects in mice [39,40]. The manual curation of these shared genes in the International Mouse Phenotyping Consortium found that *GIGYF2* related to phenotypes of tremors and increased fasting circulating glucose levels in knockout mice, and *SMG6* related to phenotypes of abnormal startle reflex and decreased hematocrit, which are risk factors for MI and SMDs. Our results suggested that identifying pleiotropic genes through meta-analyses of GWASs was biologically sensible.

The gene-set analyses by MAGMA showed nine pathways shared between MI and SMDs. These pathways can be categorized into four main groups: neurotransmitter regulation, lipid metabolism, chemical homeostasis, and neuronal development and function. GOBP regulation of catecholamine uptake involved in synaptic transmission (GO:0051940), GOBP neurotransmitter transport (GO:0006836), and GOBP regulation of neurotransmitter levels (GO:0001505) altogether showed the importance of neurotransmitters in this comorbidity. Neurotransmitters are fundamental chemical messengers in the nervous system governing chemical communication between cells and playing a vital role in physiology and physical health. Dysfunction and abnormalities in the levels of neurotransmitters play a critical role in the pathophysiology of SMDs, and there are several generations of pharmaceutical drugs designed based on neurotransmitters such as atypical antipsychotics, selective serotonin reuptake inhibitors, and serotonin-norepinephrine reuptake inhibitors [41]. In GOBP regulation of neurotransmitter levels (GO:0001505), Catechol-O-methyltransferase (*COMT*) achieved nominal significance both in BD and MI (*p* value for BD = 0.023, *p* value for MI = 4.65 × 10^−3^). *COMT* is an enzyme that plays a crucial role in the metabolism of neurotransmitters, including dopamine, norepinephrine, and epinephrine, and has often been suggested as a famous candidate gene in the development of schizophrenia. Functional COMT Val158Met polymorphism is also linked to a higher risk of acute coronary events [42]. Additionally, several significant cellular components from GOCC as gene sets indicated that lipid metabolism can exert effects on SMDs and MI such as high-density lipoprotein particle (GO:0034364), chylomicron (GO:0042627), and triglyceride-rich plasma lipoprotein particle (GO:0034385). In the high-density lipoprotein particle (GO:0034364) pathway, *APOA1* (*p* value for BD = 5.58 × 10^−4^; *p* value for MI = 0.0421), *APOA5* (*p* value for BD = 0.0187; *p* value for MI = 2.98 × 10^−6^), *APOC4* (*p* value for BD = 0.0126; *p* value for MI = 7.72 × 10^−4^), and *APOM* (*p* value for BD = 1.20 × 10^−5^; *p* value for MI = 3.07 × 10^−3^) achieved nominal significance in BD and MI, and influence MI by modulating the structure and function of high-density lipoprotein particles. Compared to healthy individuals, patients with schizophrenia exhibited a decrease in ApoA1 levels [43]. A decreased level of ApoA1 in BD patients is also reported by Song et al. and Sussulini et al. [44,45]. These research studies, together with our findings, indicate that lowered ApoA1 levels might be associated with cognitive dysfunction and neuroinflammation, and thereby contribute to the development of SMDs. Other pathways, such as long-term potentiation (PATHWAY: hsa04720) and dendritic arborization (GO:0140059), suggest that neuronal development and function may also play a significant role in the pathogenesis of SMDs and MI. These pathways may explain the interplay between SMDs and MI. An interesting result of *NRP1* identified by gene-set analyses indicated that its function may contribute to the pathophysiology of these disorders. The altered expression or function of NRP1 may affect processes such as neuronal development, synaptic plasticity, and neurovascular coupling in SCZ and BD [46]. A knockout mouse study shows that the knockout of NRP-1 in cardiomyocytes and vascular smooth muscle cells leads to the development of cardiomyopathy and causes the mice to be prone to heart failure after MI [47]. The genes identified in the GWAS analysis with nominal significance in both BD and MI of GO:0001505 and GO:0034364 pathways were listed in Appendix A.

To investigate the contribution of rare variants to the genetic basis of MI and SMD comorbidity, we performed gene-based and gene-set association analyses using UKBB WES. Only one gene (*MUC2*) achieved exome-wide significance. MUC2 is a key gel-forming glycoprotein that is abundantly secreted and constitutes the primary constituent of the mucus layer lining the gastrointestinal tract, particularly in the large intestine or colon [48]. The possible mechanism behind this result may be the role of the gut in both MI and SMDs. Alterations in MUC2 function can lead to increased intestinal permeability, which has been implicated in systemic inflammation, a factor associated with severe mental disorders through the gut–brain axis and the onset of cardiovascular diseases [49,50]. *Akkermansia muciniphila*, the mucin-degrading bacterium, can improve cardiovascular health by modulating gut inflammation [51]. From a clinical standpoint, probiotic supplementation can enhance gastrointestinal well-being, cognitive performance, and immune function, and may also have potential benefits for psychiatric symptoms [52]. Several zinc-finger protein (ZFP) genes, including *ZNF774* in BD (*p* value = 8.19 × 10^−4^), *ZNF311* in SCZ (*p* value = 5.04 × 10^−4^), and *ZNF652* in MI (*p* value = 9.67 × 10^−4^), were found to be nominally and FDR-significant in all disease tests (Appendix A). ZFPs are a family of transcription factors characterized by zinc-binding motifs that facilitate DNA recognition and binding. The dysfunction of ZFPs may contribute to disrupted neurotransmitter systems, neuronal networks, and cardiovascular development linked to SMDs and CVDs [53,54]. Zinc-finger protein gene therapy targeting VEGF-A can contribute to treating brain damage and artery disease [54,55]. As for gene-set analyses, we revealed that insulin processing, neutrophil-mediated killing of Gram-negative bacterium, androgen catabolic process, and angiotensin receptor binding were shared between multiple phenotypes, which indicated that these biological processes may play a vital role in comorbidity and have already been proven by some studies [56,57,58,59].

Our results demonstrated the value of rare functional variant calling in a case–control study design for understanding comorbidity. Notably, we performed an exome-wide association study on genes and gene sets in addition to GWAS analyses unlike prior studies [17,60]. Moreover, by selecting controls from the same cohort (i.e., UKBB) as cases, our approach allows us to identify more credible genes and gene sets contributed by rare variants with potentially larger effect sizes, compared to relying solely on allele frequency information from databases such as ExAC and gnomAD.

Some of the genes illustrated above interacted with therapeutic drugs for MI or SMDs according to the drug–gene interaction analysis. In total, we found 28 drugs targeting six genes. Our findings on these drugs showed clinical importance. Antipsychotics such as olanzapine and clozapine are the mainstay treatment for people with schizophrenia or related disorders. Both lamotrigine and carbamazepine are used in clinical practice for the treatment of bipolar disorder, particularly for managing manic or depressive episodes. The administration of ticlopidine can be part of secondary prevention strategies for patients who have experienced MI with an apparent trend toward a lower reinfarction rate [61]. A double-blind study shows that methazolamide is a potential therapy for CVDs with clinical benefits beyond glucose control [62].

The present study has several limitations. First, the sample size of WES was not large enough to detect convincing genes and pathways related to MI and SMDs. Larger sample sizes or replication studies may find that more genes and pathways achieved statistical significance. Second, we only considered genetic factors from a genomic perspective. Integrating results from other omics, such as metabolomics or proteomics, may provide new insights into these genetic underpinnings of comorbidity. Thirdly, environmental factors such as smoking, lack of exercise, and adverse metabolic effects of antipsychotic medications were not considered in our study.

## 5. Conclusions

In conclusion, our study provides a comprehensive analysis of the genetic underpinnings underlying the comorbidity between MI and SMDs by employing a combined GWAS and EWAS approach. Our findings highlight several key genes, such as *HLA-B*, *HLA-C*, *NRP1*, and *MUC2*, and pathways, such as neurotransmitter regulation and lipid metabolism, which are implicated in the development and progression of MI and SMDs. The combination of GWAS and EWAS data, along with the identification of specific genes and pathways, opens up new possibilities for future research.

Our study adds to the existing evidence that underscores the relationship between cardiovascular and mental health. It sets the stage for further exploration of the shared genetic and biological mechanisms underlying this comorbidity. By elucidating shared mechanisms among MI and SMDs, we hope to facilitate the development of integrated treatment approaches that address both the physical and mental aspects of these conditions, ultimately improving patient outcomes and quality of life.

## Figures and Tables

**Figure 1 biomedicines-12-02298-f001:**
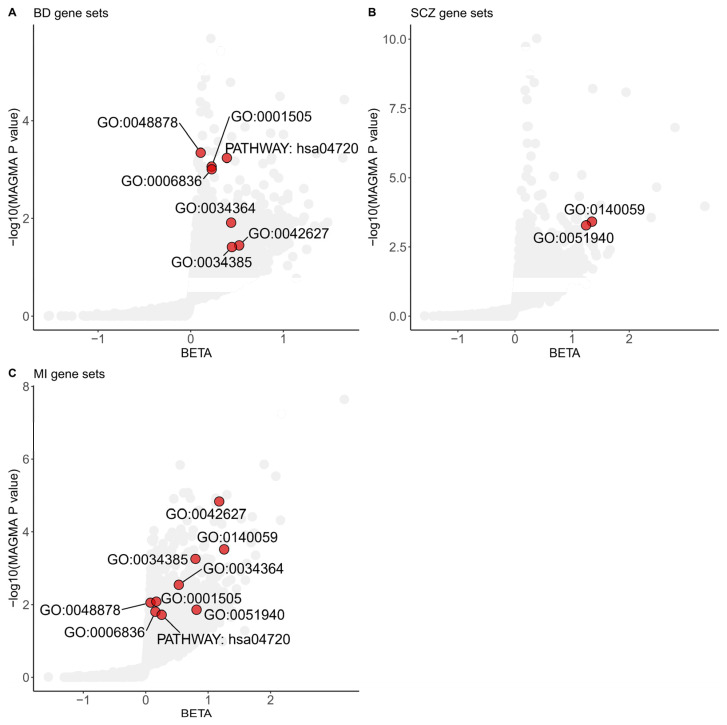
Shared gene sets among Myocardial Infarction and severe mental disorders in GWAS analyses. Gene sets enriched in MAGMA analyses of the BD (**A**), SCZ (**B**), and MI (**C**) significant common architecture are shown in the plots. The dots colored red represent significant common gene sets based on a Benjamini–Hochberg correction for multiple testing at FDR < 0.25 and nominal *p* value < 0.05. The data are available in Appendix A. The specific term names of gene sets from GO or KEGG are listed below. GO:0048878, chemical homeostasis; GO:0001505, regulation of neurotransmitter levels; GO:0006836, neurotransmitter transport; GO:0034364, high-density lipoprotein particle; GO:0042627, chylomicron; GO:0034385, triglyceride-rich plasma lipoprotein particle; GO:0140059, dendrite arborization; GO:0051940, regulation of catecholamine uptake involved in synaptic transmission; PATHWAY: hsa04720, long-term potentiation. BD: Bipolar Disorder; SCZ: Schizophrenia; MI: Myocardial Infarction.

**Figure 2 biomedicines-12-02298-f002:**
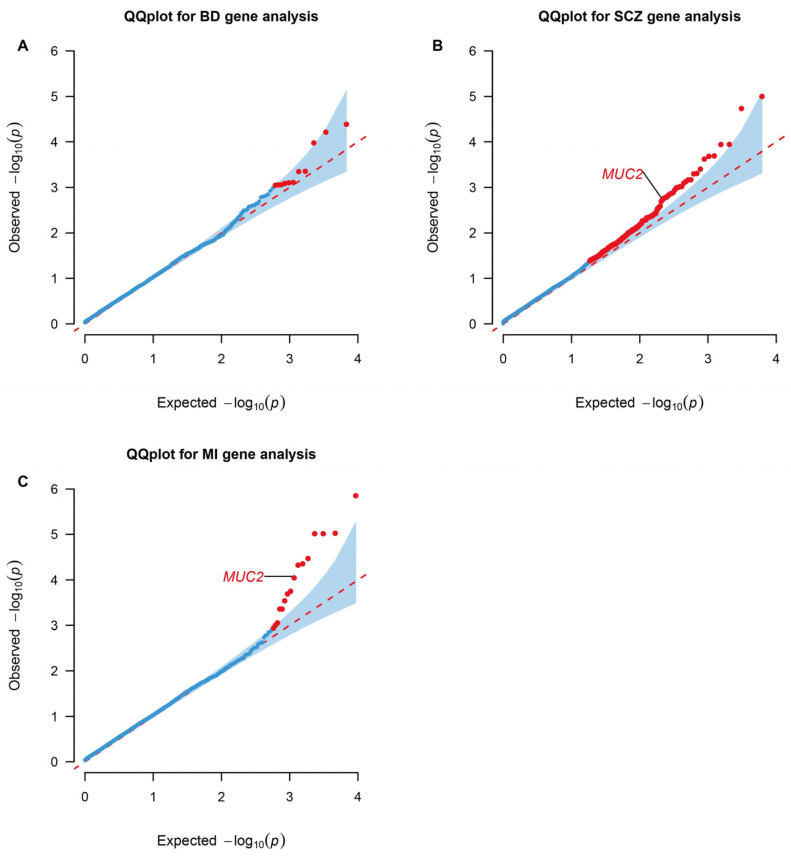
QQ plots for gene-level association tests in WES analyses. Genes enriched in RVtests analyses of the BD (**A**), SCZ (**B**), and MI (**C**) significant rare architecture are shown in the plots. The dots colored red represent statistically significant genes based on a Benjamini–Hochberg correction for multiple testing at FDR < 0.25 and *p* value < 0.05. BD: Bipolar Disorder; SCZ: Schizophrenia; MI: Myocardial Infarction. The data are available in Appendix A.

**Figure 3 biomedicines-12-02298-f003:**
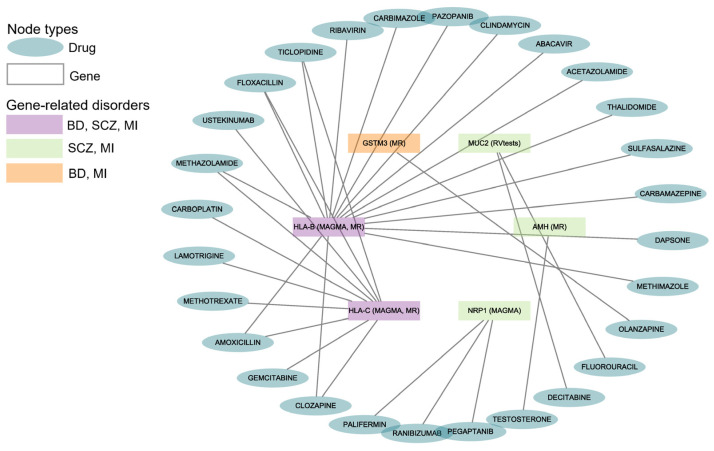
Gene–drug interactions for BD, SCZ, and MI. The diagram reveals the correlation between the disorders, genes, and drugs. BD: Bipolar Disorder; SCZ: Schizophrenia; MI: Myocardial Infarction. The data are available in Appendix A.

**Table 1 biomedicines-12-02298-t001:** Overview of the cohorts in the MAGMA analyses.

Cohort	Phenotype ^1^	N Cases/Controls	Ancestry
Psychiatric Genomics Consortium (PGC)	BD	41,917/371,549	European
Psychiatric Genomics Consortium (PGC)	SCZ	53,386/77,258	European
Cardiovascular Disease Knowledge Portal (CVDKP)	MI	17,505/454,212	European

^1^ The specific phenotype for the GWAS summary statistics. BD: Bipolar Disorder; SCZ: Schizophrenia; MI: Myocardial Infarction.

**Table 2 biomedicines-12-02298-t002:** Demographic and clinical characteristics of cases and controls in UK Biobank.

Ancestry	Characteristic	All Subjects	BD	SCZ	MI	Control
All ancestry	No.	200,632	590	322	3673	2000
	Male, No. (%)	90,154 (44.9)	242 (41.0)	195 (60.6)	2907 (79.1)	841 (42.1)
	Female, No. (%)	110,478 (55.1)	348 (59.0)	127 (39.4)	766 (20.9)	1159 (57.9)
	Age, mean (SD), y	56.5 (8.1)	55.5 (8.1)	54.0 (8.3)	61.7 (6.0)	54.8 (8.1)
European ancestry	No. (%)	118,251 (93.8)	543 (92.0)	255 (79.2)	3453 (94.0)	1890 (94.5)
	Male, No. (%)	84,515 (44.9)	227 (41.8)	158 (62.0)	2736 (79.2)	787 (41.6)
	Female, No. (%)	103,736 (55.1)	316 (58.2)	97 (38.0)	717 (20.8)	1103 (58.4)
	Age, mean (SD), y	56.7 (8.0)	55.8 (8.0)	54.8 (8.1)	61.9 (5.8)	55.1 (8.0)
Others	No. (%)	12,368 (6.2)	47 (8.0)	67 (20.8)	220 (6.0)	110 (5.5)

BD: Bipolar Disorder; SCZ: Schizophrenia; MI: Myocardial Infarction.

**Table 3 biomedicines-12-02298-t003:** Shared genes among Myocardial Infarction and severe mental disorders in GWAS analyses.

Gene	MI *p* Value	BD *p* Value	SCZ *p* Value
*GIGYF2*	1.04 × 10^−6^	ns	2.80 × 10^−11^
*KCNJ13*	1.34 × 10^−6^	ns	3.05 × 10^−14^
*PCCB*	1.37 × 10^−7^	ns	1.90 × 10^−9^
*STAG1*	1.96 × 10^−6^	ns	2.01 × 10^−7^
*HLA-C*	9.56 × 10^−7^	1.68 × 10^−6^	1.82 × 10^−10^
*HLA-B*	4.98 × 10^−7^	3.24 × 10^−10^	2.45 × 10^−18^
*FURIN*	5.00 × 10^−10^	4.03 × 10^−7^	5.22 × 10^−15^
*FES*	5.00 × 10^−10^	ns	6.94 × 10^−14^
*SMG6*	6.51 × 10^−13^	ns	1.51 × 10^−7^

ns = Not significant at a Bonferroni corrected *p* value. BD: Bipolar Disorder; SCZ: Schizophrenia; MI: Myocardial Infarction.

## Data Availability

The original contributions presented in the study are included in the article/Appendix A, further inquiries can be directed to the corresponding authors.

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
