# Peer review of "Genome-Wide and Exome-Wide Association Study Identifies Genetic Underpinning of Comorbidity between Myocardial Infarction and Severe Mental Disorders†"

_biomedicines, 2024, doi:10.3390/biomedicines12102298_

Round 1

Reviewer 1 Report

Comments and Suggestions for Authors

The present study investigated shared rare and common genetic underpinning in patients with myocardial infarction, schizophrenia and bipolar disorders, based on the data from the Psychiatric Genomics Consortium (and the Cardiovascular Disease Knowledge Portal.

Identified were nine pleiotropic genes shared between SCZ and MI, and three pleiotropic genes shared between MI and BD. 221.Among those genes, three genes (HLA-C, HLA-B, and FURIN) were identified to have shared effect. This is interesting and important study, given the frequent comorbidity between the CVD and SMD.

I have only few minor issues:

„...showed an increase in mortality from cardiovascular and cerebrovascular disease [9]...” compared to which population?

It is mentioned that „All samples of cohorts were individuals of European ancestry”, but in table 2 in row 1, there is both „All ancestry” and „European ancestry”, so please explain

Line 230-”...from four different diseases or disorders...”, please provide data which four disorders, because BD, MI and SCH were studied

Reviewer 2 Report

Comments and Suggestions for Authors

The manuscript is well presented and needs only minor corrections, mainly in the presentation of the data. 

1) Please consider a revision of figure 1 - the description is too small and it should be self- explaining. 

2) In the discussion the interference between the psychiatric disorders and cardiovascular diseases should be more clearly stated. 

3) What are the implications of the association of MUC2 ? Please discuss. 

Comments on the Quality of English Language

Only minor changes are required. 

Reviewer 3 Report

Comments and Suggestions for Authors

Manuscript submitted in Biomedicines entitled “Genome-wide and Exome-wide Association Study Identifies 2 Genetic Underpinning of the Comorbidity Between Myocardial 3 Infarction and Severe Mental Disorders” by Jiang and coauthors is an interesting study that might contribute to research domain. However, there are several concerns that need to be addressed first before further consideration for publication in Biomedicines.

Please revise first para of introduction.

Page 2 lines 58-70, showing extensive plagiarized text which is not appreciable, please rewrite.

It would be better if author just use bipolar disorders and schizophrenia, rather than using several mental disorders in multiple instances.

What is the reason behind choosing only bipolar disorder and Schizophrenia?

Please do not use statements with acronyms. Throughout the manuscript such mistakes are present.

Try to simplify figure 3, if possible.

Write a separate section for statistical analysis and significance levels wherever applicable. 

Second paragraph of discussion is not clear, rewrite.

Discuss nine pathways that shared between MI and SMDs for better understanding and rational of their targeted gene chosen in the study.

It would be better to elaborate conclusion.

There is ambiguity in the references and some places typos are present therefor authors are advised overcome such mistakes in revised paper.

Reviewer 4 Report

Comments and Suggestions for Authors

In this interesting work, Authors addressed the identification of genes and pathways shared by common disease like myocrdial infarction, skizophrenia e bipolar disorder. Results are interesting and intriguing.

My only comment concerns the concept that a disregulation affecting a gene or a gene set plays a similar role in different diseases investigated. Authors should also discuss the hypothesis that a specific gene or gene set can act not in the same direction (i.e. downregulation) in the different diseases. There is room for a role for gain-of-function variants?
